# Aeroacoustic Optimization Design of the Middle and Upper Part of Pantograph

Jing Guo [1], Xiao-Ming Tan [1,2,3,*] , Zhi-Gang Yang [3], Yu-Qi Xue [1], Ya-Nan Shen [1] and Hao-Wei Wang [1]

1   College of Mechanical Engineering Hunan Institute of Science Technology, Yueyang 4140083, China
2   Key Laboratory of Aerodynamic Noise Control, China Aerodynamics Research and Development Center, Mianyang 621000, China
3   Key Laboratory of Traffic Safety on Track, Ministry of Education, School of Traffic & Transportation Engineering, Central South University, Changsha 410075, China
*   Correspondence: tanxiaoming_csu@163.com; Tel.: +86-15274972406

**Abstract:** The pantograph is the main noise source of high-speed trains, of which the middle and upper parts of the pantograph account for about 50% of the whole noise energy. Taking CRH380BL pantograph as the basic prototype, three aerodynamic noise reduction measures of opening, slotting, and airfoil are introduced to build a new pantograph, and their aeroacoustic performances are comprehensively investigated through large eddy simulation (LES) and Ffowcs Williams–Hawkings (FW-H) equation method. The research results show that the open upper and lower arms (ULA) can reduce the downstream vorticity intensity and vortex structure scale, which in turn reduces the noise source intensity, thus reducing their radiated noise by approximately 1.1 dBA. The slotted ULA reduce the size of the rear vortex structure but increase the vorticity intensity, so it is difficult to effectively control their radiated noise. The airfoil bow head reduces the vorticity intensity and vortex structure scale behind it, and avoids periodic vortex shedding, thereby reducing its noise source intensity, thus reducing its radiated noise by about 1.2 dBA.

**Keywords:** pantograph; aerodynamic noise; large eddy simulation; optimal design

## 1. Introduction

Existing research shows that when the train speed is greater than 300 km/h, the proportion of aerodynamic noise source energy to total noise source energy is about 50% [1–3]. The pantograph is one of the main aerodynamic noise sources. Pantographs are composed of bars of different scales and inclination angles, which induce complex flow field phenomena such as layer/turbulence boundary layer separation, vortex streets, flow transitions, and large separation turbulence, and become significant noise sources [4,5].

In terms of pantograph region aerodynamic noise characteristics and generation mechanism, the acoustic wind tunnel test results of Andreas Lauterbach and others [5] show that the pantograph has obvious wind-blown sound characteristics, and its far-field radiated noise has Strohal number similarity. Siano et al. [6] calculated the closed pantograph aerodynamic noise for 300 km/h using SAS and acoustic boundary elements and pointed out that vortex shedding is the main formation mechanism of pantograph aerodynamic noise. Sun et al. [7] and Lee et al. [8] both analyzed the aerodynamic noise contribution rate of pantograph components and obtained some accurate conclusions, for example, the high-frequency noise mainly originated from the bow head. Holmes et al. [9] studied the sound generation mechanism of the external convex cavity deflector around the pantograph and found that the transverse vortex dislodged from the cavity guide edge hit the cavity with the edge and generated a strong fluctuating pressure inside the cavity. In terms of pantograph aerodynamic noise optimization measures, the main focus is on optimizing the rod aerodynamic shape, adding deflector structures, etc. Kurita T. [10] et al. explored in detail the acoustic characteristics of the guide edge shape, its cross-sectional

shape, and the inclination angle of the pantograph deflector, and designed a new low-noise deflector with a decreasing 2 dB. Takaishi et al. [11] and Ikeda et al. [12,13] all pointed out that suitable bow-head struts can destroy its vortex street, thus effectively suppressing its radiated noise. Zhang Ya-dong et al. [14] showed that the noise radiated from an open pantograph is 3.4 dBA lower than that of a closed one. Rho et al. [15] used the LES, FW-H equation, and genetic algorithm-based Kriging model to optimize the cross-sectional shape of the bow head, and the optimal cross-sectional shape of the bow head with optimal aerodynamic/acoustic characteristics was obtained. The optimized effect of 26% drag reduction and 2 dB noise reduction was achieved. Liu Xiao-wan et al. [16] numerically investigated the flow/acoustic field characteristics of a cylindrical rod with different angles of attack and incoming flow velocities and found that the higher the angle of attack, the lower the peak sound pressure level and frequency of aerodynamic noise. Sun et al. [7] pointed out that optimizing the shape of the deflector can effectively suppress the cavity radiation noise. Kim et al. [17] simplified the pantograph and sinking platform into a cylinder and a concave cavity, respectively, discussed the influence of the position of the cylinder installation and the rounding angle of the guide edge on the aerodynamic noise and obtained the following conclusions: the backflow length of the cavity and the vortex are significantly reduced during the growth of the circular angle from 0 to a certain angle, and the total radiated aerodynamic noise, total resistance and total lateral force are also significantly reduced; increasing the distance between the guide edge and the cylinder can effectively suppress the mutual interference behavior between the shear flow and the cylinder, thus reducing the peak sound pressure level.

At present, the research on the mechanism of aerodynamic noise in the pantograph area has a relatively clear understanding, and the measures mainly focus on adding deflectors on both sides of the pantograph to isolate the noise radiation, or installing the pantograph in the cavity to reduce the incoming speed at the bottom, or carrying out aerodynamic noise reduction design for the shape of the rod [18–20], or laying sound-absorbing materials on the back of the arm to achieve noise reduction, with less discussion of the applicability of various aerodynamic noise reduction design methods in the pantograph area such as openings, slotting and airfoil.

The middle and upper part of the pantograph is composed of typical rod structures such as the upper arm rod, lower arm rod, and bow head, which can become a strong aerodynamic noise source under high-speed airflow, and is located at a higher position, making it difficult to take effective measures to cut off the noise propagation. 330 km/h speed class real vehicle tests show that the contribution of the pantograph to the radiated noise of high-speed trains is more than 10% [21], among which the contribution of the middle and upper part of the pantograph to its radiated noise is more than 50%. Therefore, this paper takes CRH380BL as the base model and adopts three types of aerodynamic noise reduction designs for it, namely, opened, slotted and airfoiled, respectively, to determine their applicability for ULA according to their respective noise reduction principles, and noise reduction magnitudes, and then obtain a new pantograph with better aerodynamic acoustic performance.

## 2. Aerodynamic Noise Optimization

The complex geometric characteristics of the pantograph make its aerodynamic noise mechanism to be complicated. Existing studies show that the pantograph flow field structure includes: the strong vorticity flow, the vortex street, and the different scale single-legged hairpin vortex [21].

These flow structures originate from the rod. Therefore, the noise reduction for the rods can effectively reduce the noise in this region. The principle of noise reduction for the rod includes: restraining the separation of airflow; reducing the vortex shedding intensity and adjusting the vortex shedding frequency; reducing the mixing intensity of airflow; restraining the mutual interference strength of members. Considering the

geometric and flow field structural characteristics of ULA, this paper adopts three noise reduction measures of opening, slotting, and airfoil to reduce its noise, as shown in Figure 1.

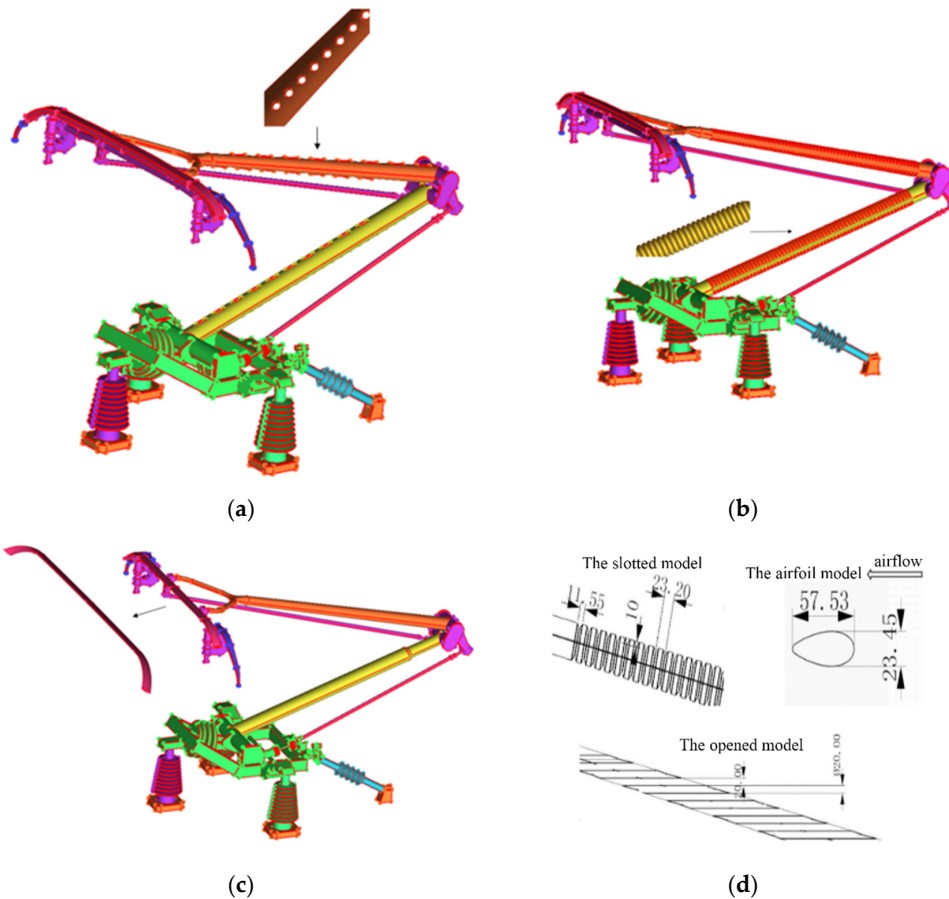

(a)                                                                                                                        (b)

(c)                                                                                                                        (d)

**Figure 1.** Schematic diagram of the optimized scheme in the upper part of the pantograph. (**a**) Opened ULA. (**b**) Slotted ULA. (**c**) The airfoil bow head. (**d**) Dimension description.

Figure 1a shows the opened ULA. The hole diameter is 20 mm, the vertical spacing between different holes is 20 mm, and the holes are horizontally oriented. There are 8 holes in the upper arm and 10 holes in the lower arm.

Figure 1b shows the slotted ULA. The groove is 11.55 mm wide and 10 mm deep, and the edge width between grooves is 23.2 mm

Figure 1c shows the airfoil bow. Its section is 57.33 mm long and 23.45 mm high.

## 3. Numerical Calculation Model

When trains are operated under open line conditions, the actual running state of the train on the ground is usually simulated by relative motion. The calculation domain is shown in Figure 2. The size of the calculation area should ensure that the flow field is fully developed. The upstream of the flow field should be no less than 8 times the characteristic height, and the downstream of the flow field should be no less than 16 times the characteristic height. The characteristic height here refers to the distance of the top surface of the train from the ground. These parameters can significantly reduce the influence of the boundary on the flow field. The size of the calculation area used in this study is 400 × 30 × 20 m, which fully meets the requirements. The boundary conditions taken by each surface of the calculation domain are shown in Table 1.

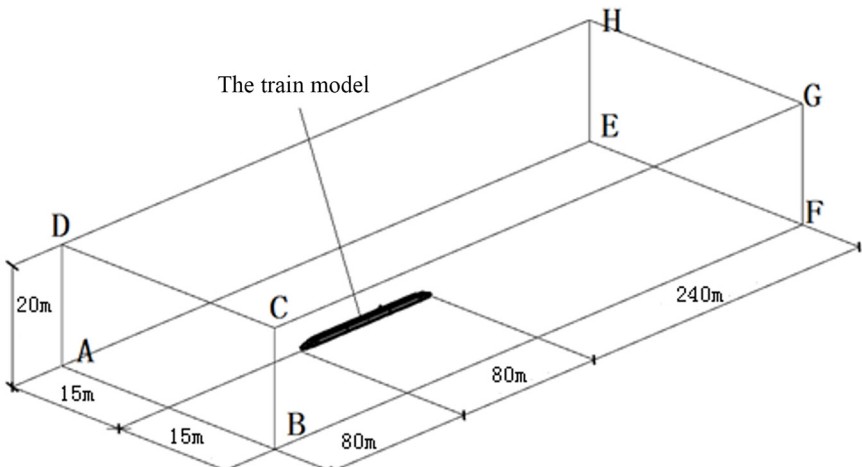

**Figure 2.** Train calculation domain.

**Table 1.** The facets of the computational domain.

| Face ABCD | Face BFGC, AEHD, CGHD | Face ABFE | Face EFGH |
|-----------|----------------------|-----------|-----------|
| Speed inlet | Symmetric boundaries | Sliding Ground | Pressure outlet |

The numerical simulation calculation of the train flow field is divided into steady-state and unsteady-state calculations. The steady-state calculation is based on the pressure-based implicit solution method. $SST - k\omega$ model is chosen for the turbulence model. The SIMPLE algorithm is used for the pressure-velocity coupling, and standard discrete format is used for the pressure, and the second-order windward discrete format is used for the momentum, turbulent kinetic energy, and turbulent kinetic energy dissipation rate. For the unsteady state calculation, the steady-state flow field is used as the initial flow field, the turbulence model is LES, and the sub-grid model is the Smagorinsky model. The PISO method is chosen for pressure-velocity coupling. The time step is taken as $5 \times 10^{-5}$ s, and a total of 10,000 time steps are calculated. All calculations were completed in Wuxi Supercomputing Center. The calculation software is ANSYS fluent. For further details of the numerical model, see articles (Vortex structures and aeroacoustic performance of the flow field of the pantograph' and 'Adaptability of Turbulence Models for Pantograph Aerodynamic noise simulation').

The calculation conditions are shown in Table 2.

**Table 2.** Calculated working conditions.

| Computational Models | Scaling Ratio | Vehicle Speed |
|---------------------|---------------|---------------|
| Opened Slotted Airfoil | 1:8 | 350 km/h |

The grid schematic is shown in Figure 3.

The calculation model grid parameters are shown in Table 3. In Table 3, $y^+$ is calculated by Equation (1).

$$y^+ = \frac{\overline{U^*}y}{v} \tag{1}$$

where $\overline{U^*} = \sqrt{\tau_\omega/\rho}$ is the wall friction velocity, $\tau_\omega$ is the wall shear stress, $y$ is the distance from the centroid of the first layer of mesh to the wall, $v$ is the kinematic viscosity coefficient.

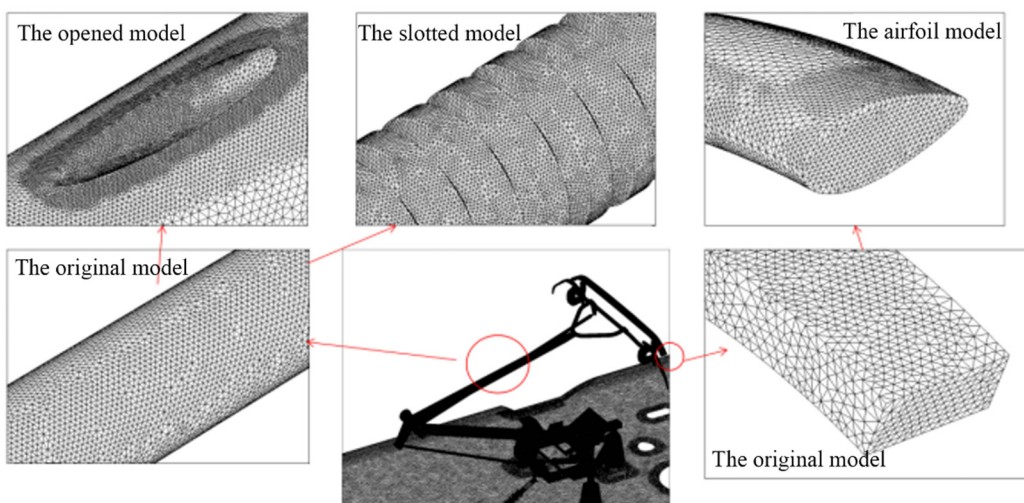

**Figure 3.** Schematic diagram of the calculation grid.

**Table 3.** Table of calculation grid parameters.

| Models | Volume Grid Number (Billion) | Total Surface Grid Number (Million) | Pantograph Surface Grid Number (Million) | Whole Vehicle Surface $y^+$ Average Value | Pantograph Surface $y^+$ Average Value |
|---|---|---|---|---|---|
| Opened | 1.91 | 1365 | 313 | 0.92 | 0.85 |
| Slotted | 1.92 | 1298 | 326 | 0.91 | 0.82 |
| Airfoil | 1.91 | 1325 | 309 | 0.93 | 0.82 |

According to Table 3, $y^+$ of the car body and the pantograph are both less than 1. This indicates that the grid can describe the airflow behavior of the viscous bottom layer. This is also an important source of aerodynamic noise. Therefore, the grid in this paper can be used to calculate the aerodynamic noise in the pantograph area.

## 4. Flow Field Fluctuation Performance

Figures 4–6 compare the original model and the optimized model, showing the velocity amplitude, vorticity amplitude, and Q-value distribution clouds on the longitudinal symmetry surface, respectively.

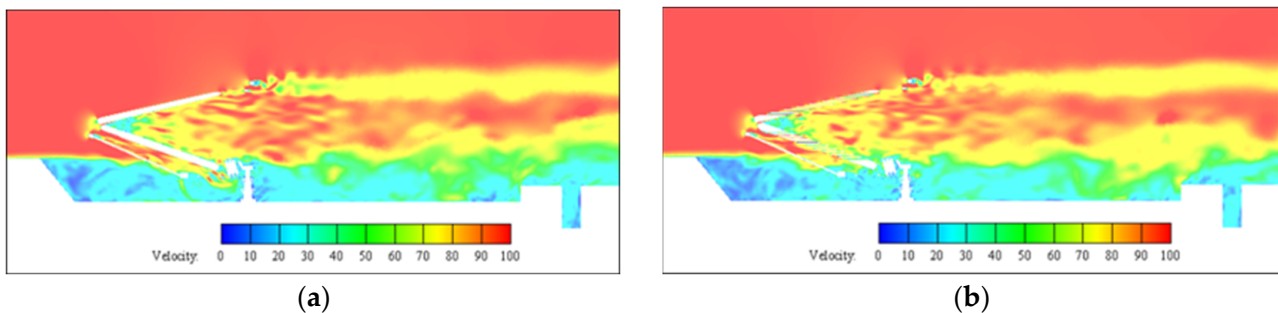

(**a**)                                                  (**b**)

**Figure 4.** *Cont.*

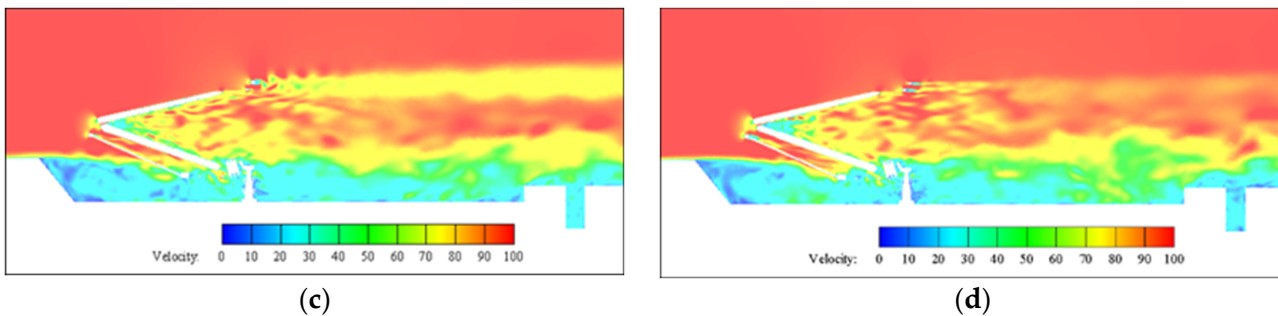

**Figure 4.** The velocity amplitude distribution on the longitudinal symmetry surface. (**a**) Original model. (**b**) Opened model. (**c**) Slotted model. (**d**) Airfoil model.

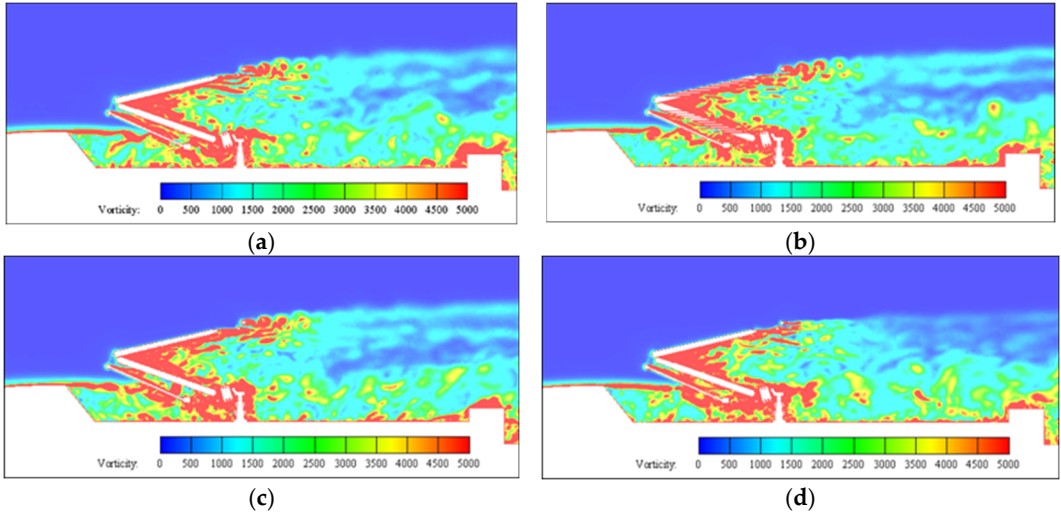

**Figure 5.** The vorticity amplitude distribution on the longitudinal symmetry surface. (**a**) Original model. (**b**) Opened model. (**c**) Slotted model. (**d**) Airfoil model.

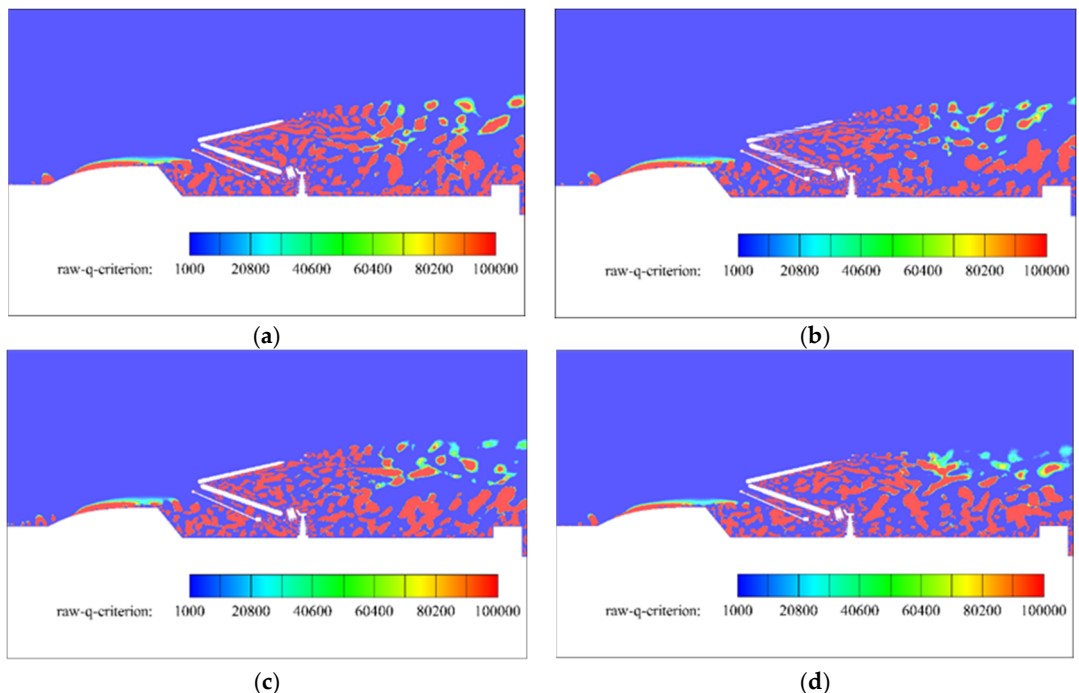

**Figure 6.** Q-value distribution on the longitudinal symmetry surface. (**a**) Original model. (**b**) Opened model. (**c**) Slotted model. (**d**) Airfoil model.

According to the original model in Figures 4–6, it can be seen that the middle and upper regions of the pantograph have typical flow field structural characteristics: the middle of the pantograph is located in the strong vortex shear flow separated by the convex plate, which makes the vorticity amplitude around this region larger, and thus changes the periodic vortex shedding characteristics of the existing flow field in this region; the upstream of the upper part of the pantograph is a high-speed low vorticity region, while the downstream is a low-speed strong vorticity region, and shows periodic vortex shedding.

According to Figures 4–6, the optimized model with respect to the original model leads to the following conclusions:

(1)    The opened ULA increase their rear airflow velocity and their upstream rear vorticity intensity, but reduce their downstream rear vorticity intensity and their rear vorticity structure scale;
(2)    The slotted ULA increase the vorticity intensity around them;
(3)    The airfoil bow head reduces the intensity of the vorticity behind the bow head and the vortex structure size, making the flow field smoother.

## 5. Aerodynamic Noise Source Performance

The time root mean square of the fluctuating pressure on the pantograph surface can reflect its sound source intensity [21]. Equation (2) is the calculation of fluctuating pressure. Equation (3) is the calculation of the time root mean square of fluctuating pressure.

$$p' = p - \overline{p} \tag{2}$$

where $p$ is the transient pressure, $p'$ is the fluctuating pressure, $\overline{p}$ is the time average of the transient.

$$\mathrm{d}pdt-\mathrm{rms} = \sqrt{\frac{\int\limits_T (p')^2 dt}{T}} \tag{3}$$

where $T$ is the sample time.

Figure 7 qualitatively compares the $\mathrm{d}pdt-\mathrm{rms}$ distribution around the pantograph between the original model, the opened model, the slotted model, and the airfoil model.

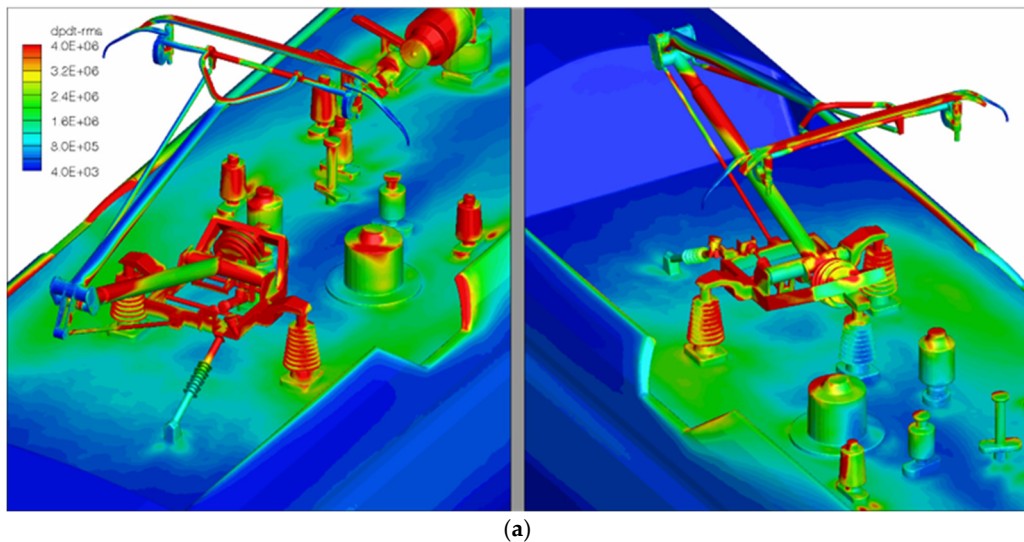

(a)

**Figure 7.** *Cont.*

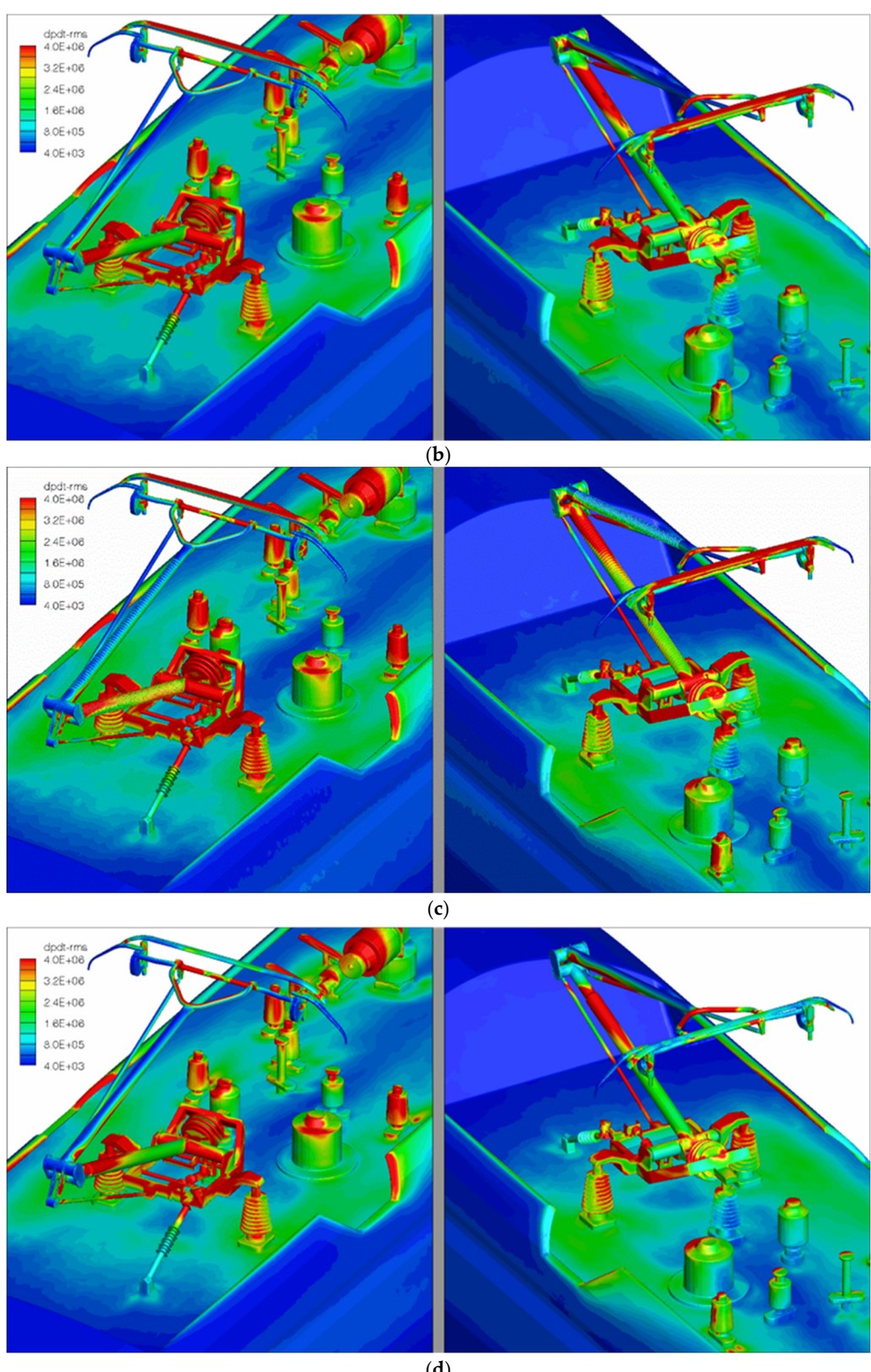

**Figure 7.** The d$p$d$t$−rms distribution cloud map around the pantograph. (**a**) Original model. (**b**) Opened model. (**c**) Slotted model. (**d**) Airfoil model.

According to Figure 7, compared with the original model, the following conclusions can be drawn:

(1) The opened ULA lightly increases the noise source intensity on the windward side, but significantly reduces the noise source intensity on the leeward side;
(2) The slotted ULA both increases the noise source intensity on the windward and leeward sides;
(3) The airfoil model significantly reduces the noise source intensity and the distribution area of the strong noise source on the bow head.

Equation (4) is the sound power calculation formula of an equivalent sound source [21].

$$W_{source} \propto \overline{\left( \int_S \frac{\partial}{\partial t} p(\mathbf{y}) \mathrm{d}S(\mathbf{y}) \right)^2} = \overline{\left( \frac{\partial}{\partial t} \int_S p(\mathbf{y}) \mathrm{d}S(\mathbf{y}) \right)^2} \tag{4}$$

In this equation: $W_{source}$ denotes the equivalent sound power of the sound source; "$\mathbf{y}$" is the vector of the sound source; $\frac{\partial p(\mathbf{y})}{\partial t}$ is the train surface fluctuation pressure time gradient; "$S$" is the noise source area.

Figure 8 shows the proportion of the three improved models relative to the original model.

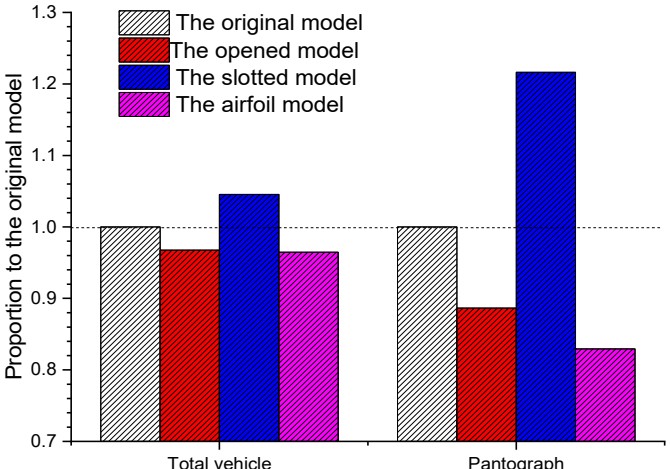

**Figure 8.** Histogram of equivalent sound power comparison between optimized solutions in the upper and middle parts of the pantograph.

According to Figure 8, the opened model and the airfoil model can effectively reduce the equivalent sound power of the pantograph region and weakly reduce the equivalent sound power of the whole vehicle; the slotted model significantly increases the equivalent sound power in the pantograph region, while increasing the equivalent sound power of the whole vehicle.

## 6. Far-Field Radiated Noise Performance Evaluation

In order to investigate the far-field noise, the receiver points are arranged longitudinally along the train. They are 25 m away from the central axis of the train and 3.5 m high from the ground. The arrangement range of the receiver points is [0 105] m, and the adjacent receiver points are separated by 7 m. The receiver points were arranged as shown in Figure 9. As can be seen from Figure 9, the first receiver point is located at the nose-tip of the train, and the other receiver points are numbered in the order of flow. The train is located between receiver point 1 and receiver point 13, and the pantographs are located between receiver points 7 and 9, respectively.

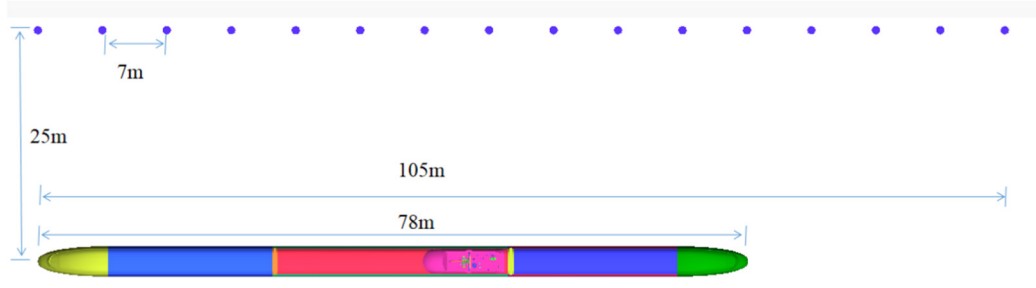

**Figure 9.** Sound receiver point layout diagram.

Figure 10 quantitatively compares the sound pressure level distribution curves between them. Compared with the original model, the opened model and the airfoil model can reduce the far-field noise of the whole vehicle and the pantograph, especially for the receiver points near the pantograph, however, the slotted model increases the far-field noise of the whole vehicle and the pantograph. This is because the opening can increase the airflow through the rod, thereby increasing the air velocity in the leeward area of the rod, and reducing the pressure in the leeward area of the rod, so that the airflow separation is backward. Delayed airflow separation can significantly reduce the intensity of aerodynamic excitation in this area, and then reduce the intensity of sound sources and the distribution area of strong sound sources.

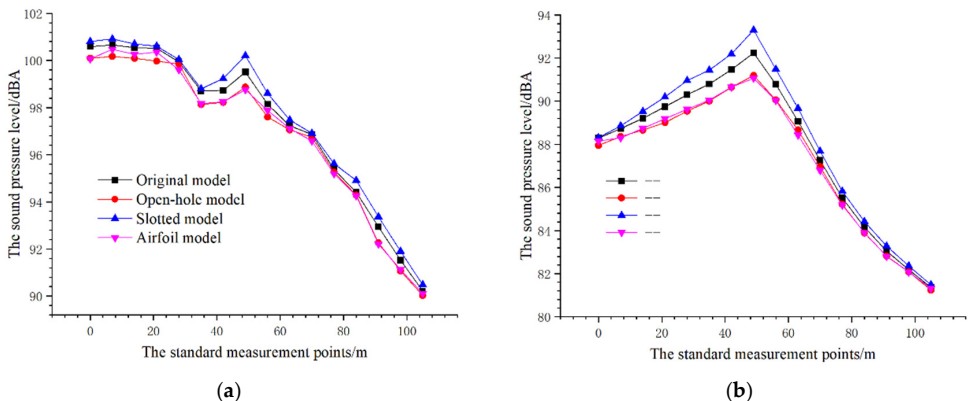

**Figure 10.** The sound pressure level distribution curves. (**a**) Complete vehicle. (**b**) Pantograph.

In summary, both the opened model and the airfoil model can effectively suppress the aerodynamic noise of the middle and upper part of the pantograph, which can reduce 1.1 dBA, 1.2 dBA of the pantograph region, and can reduce 0.7 dBA, 0.8 dBA of the whole vehicle, respectively. The slotted model is difficult to effectively control the aerodynamic noise around the pantograph.

## 7. Conclusions

The opened ULA can suppress the aerodynamic noise in the upper and middle regions of the pantograph, which can reduce the far-field noise of the pantograph region by about 1.1 dBA, or can reduce the far-field noise of the whole vehicle by about 0.7 dBA.

The airfoil bow head can also suppress the aerodynamic noise in the upper and middle regions of the pantograph, which can reduce the far-field noise in the pantograph region by about 1.2 dBA, or reduce the far-field noise of the whole vehicle by about 0.8 dBA.

Slotted ULA enhances the aerodynamic noise in the pantograph mid-upper region.

It should be noted that optimizing the location and scale of the opened ULA and the airfoil bow head can further suppress the airflow interference of various components, so as to obtain the maximum noise reduction effect.

**Author Contributions:** Conceptualization, Z.-G.Y.; Data curation, Y.-Q.X.; Formal analysis, Y.-N.S.; Methodology, X.-M.T.; Software, H.-W.W.; Writing—original draft, J.G. All authors have read and agreed to the published version of the manuscript.

**Funding:** Projects(ANCL20200302) supported by Key Laboratory of Aerodynamic Noise Control.

**Institutional Review Board Statement:** Hunan Institute of Science and Technology.

**Informed Consent Statement:** Not applicable.

**Conflicts of Interest:** The authors declare no conflict of interest.

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
