# Peer review of "Aeroacoustic Optimization Design of the Middle and Upper Part of Pantograph"

_applsci, doi:10.3390/app12178704_

Round 1
Reviewer 1 Report
The article seems very interesting and the conclusion from its results are followed logically. However, the Authors need to explain in detail how the results were produced. Which numerical models were used? Is this an in-house code or a commercial one? Chapter 3 is too short and does not contain all the appropriate information. Finally, they need to show a validation case for the numerical code they used, in order to prove the reliability of their results to the reader.
Attached you will find a pdf file with my comments.
P.S.: It is more convenient to have the draft manuscript with line numbering. This way, the reviewer can link the corrections with the number of the text line.

Author Response
The software used in this paper is ANSYS fluent. Since this is a continuation of the previous articles published by the author, the author briefly describes the numerical model and model validation. These articles include ‘Vortex structures and aeroacoustic performance of the flow field of the pantograph’ and ‘Adaptability of Turbulence Models for Pantograph Aerodynamic noise simulation’ .
Reviewer 2 Report
A section on Verification and Validation of the results should be added and some descriptions on justification of accuracy of the results should be included. Moreover, a thorough revision is needed to rectify quite a lot of grammatical mistakes. Finally, presentation of the results should significantly be improved. Below are further details of some examples of the required presentation modifications:
- The quality of figures should be improved. For example, Figure 1 should be enlarged to show better the details of the sub-figures. Moreover, an appropriate descriptions for the sub-figures, should be added to the caption.
- Page 3, line 4; "The upstream of the flow field should be...": Appropriate reference and justification for the referred lengths should be provided.
- Page 5, second para: "optimization model" should be changed to "Optimised model".
- Page 6, Eq. 1: Where does the equation come from? Any ref.?
Author Response
The software used in this paper is ANSYS fluent. Since this is a continuation of the previous articles published by the author, the author briefly describes the numerical model and model validation. These articles include ‘Vortex structures and aeroacoustic performance of the flow field of the pantograph’ and ‘Adaptability of Turbulence Models for Pantograph Aerodynamic noise simulation’ . The author systematically revised the grammatical errors in the paper and marked them with yellow.
(1) The author replaced all the poor quality pictures with high quality ones.
(2) I added this sentence to the paper: These parameters can significantly reduce the influence of the boundary on the flow field.
(3) I have modified it and marked it yellow.
(4) This equation is from reference 21. the author added a reference in the revised manuscript.
Round 2
Reviewer 1 Report
The article is improved. Some minor comments are highlighted on the attached pdf file.
For next submissions: Please be sure that the draft manuscript has line numbering!

Author Response
I systematically revised the full paper in strict accordance with the requirements of the reviewers and marked it in purple. This includes blank rows before and after the table, adding statements, modifying icons, and so on.
Reviewer 2 Report
The authors have addressed my comments.
Author Response
I improved the language of the paper.